# Sol-Gel Synthesis and Characterization of Novel Y$_{3-x}$M$_x$Al$_{5-y}$V$_y$O$_{12}$ (M—Na, K) Garnet-Type Compounds

Diana Vistorskaja [1], Andrius Laurikenas [1], Alejandro Montejo de Luna [1,2], Aleksej Zarkov [1], Sapargali Pazylbek [3] and Aivaras Kareiva [1,*]

[1] Department of Inorganic Chemistry, Vilnius University, Naugarduko 24, LT-03225 Vilnius, Lithuania
[2] Faculty of Science, Universidad de Cordoba, Ctra. Madrid-Cádiz Km. 396. 14071 Cordoba, Spain
[3] Department of Physics, South Kazakhstan State Pedagogical University, 13 A Baitursynov St, Shymkent 160005, Kazakhstan
[*] Correspondence: aivaras.kareiva@chgf.vu.lt

**Abstract:** In this study, for the first time to the best of our knowledge, the new garnets Y$_{3-x}$Na$_x$Al$_5$O$_{12}$, Y$_{3-x}$K$_x$Al$_5$O$_{12}$, Y$_3$Al$_{5-y}$V$_y$O$_{12}$, and Y$_{3-x}$Na$_x$Al$_{5-y}$V$_y$O$_{12}$ with various stoichiometric compositions were successfully synthesized by the aqueous sol-gel method. All obtained samples were characterized by X-ray diffraction (XRD) analysis, Fourier-transform infrared spectroscopy (FTIR), scanning electron microscopy (SEM), and energy-dispersive X-ray spectroscopy (EDX). It was determined from the XRD results that the formation of monophasic Y$_{3-x}$Na$_x$Al$_5$O$_{12}$, Y$_{3-x}$K$_x$Al$_5$O$_{12}$, Y$_3$Al$_{5-y}$V$_y$O$_{12}$, and Y$_{3-x}$Na$_x$Al$_{5-y}$V$_y$O$_{12}$ garnets is possible only at limited doping levels. The highest substitutional level of doped metal was observed for the YAG doped with sodium (x = 1), and the lowest substitutional level was observed for the YAG doped with vanadium (y = 0.05). Furthermore, the obtained FTIR spectroscopy results were in good agreement with the XRD analysis data, i.e., they confirmed that the YAG is the main crystalline phase in the end products. The SEM was used to study the morphology of the garnets, and the results obtained showed that all synthesized samples were composed of nano-sized agglomerated crystallites.

**Keywords:** YAG; sodium; potassium; vanadium; substitution effects; sol-gel synthesis

## 1. Introduction

Garnet is the common name of minerals that are abundant in the Earth's crust and upper mantle [1]. The synthetic garnet compounds are well recognized as an important class of materials for advanced optical technologies [2,3]. The garnets doped with lanthanide or transition elements are widely used in light-emitting diodes [2,4,5], cathode-ray tubes (CRTs) [6], fiber-optic telecommunication systems [7], scintillators [8], and electroluminescent displays [3,9]. These applications are intimately related to the unique physical and chemical properties of garnet crystals [6,7,9–11].

The yttrium aluminum garnet (Y$_3$Al$_5$O$_{12}$, YAG) is an advanced synthetic garnet that has received a huge interest in studies for various kinds of applications [12,13]. The YAG is best known as a host material in solid-state lasers due to its optical transparency from ultraviolet to infrared [4,8,14–16]. For example, YAG:Nd is acknowledged as a laser gain medium, and it can be used in cosmetic surgery and in manufacturing for engraving and etching various metals and plastics [17–19]. It is well established that cerium-doped YAG (YAG:Ce) is a perfect yellow-emitting component for white light-emitting diodes (wLEDs) [7,20–23]. Furthermore, YAG is a promising candidate for high-temperature applications as a thermal barrier coating (TBCs) due to its superior high-temperature mechanical properties, low creep at high temperatures, and excellent thermal and chemical stabilities [7,24]. Additionally, YAG doped with transition metal ions (Cr$^{4+}$, Co$^{2+}$, and V$^{3+}$) is especially attractive for passive Q-switching crystals, which are used in medicine and fiber-optic telecommunication systems [25,26].

The choice of the synthesis method is a very important task because this is dependent on such powder characteristics as size, shape, the level of agglomeration, and phase purity [6,7]. Traditionally, garnets are synthesized by conventional solid-state reactions using metal oxides [5,27]. In addition, by this method, pure YAG can be obtained only at high calcination temperatures (>1600 °C) with a long heating period. Moreover, intermediate phases, such as $YAlO_3$ (YAP) and $Y_4Al_2O_9$ (YAM), are often found in the final products due to insufficient mixing and the low reactivity of the starting materials [9,28]. In order to avoid these drawbacks of the solid-state reaction [28,29], low-temperature synthesis methods such as solvothermal [21], co-precipitation [16], and sol-gel [29,30] have been developed. The sol-gel method does not require long heat treatments, high temperatures, or high pressure. Further, this synthesis method has the advantages of single-phase particles, high reactivity of raw materials, and excellent chemical homogeneity of the final product. Thus, the sol-gel method is widely used to prepare garnet-type materials [6,9,10,26,30–32].

The engineering compositional disorder in crystalline materials becomes a powerful tool to improve their properties and achieve new areas of application [2,7,33]. In this work, the yttrium aluminum garnet matrix was modified by doping with different amounts of sodium, potassium, and vanadium. Therefore, the aim of this study was to synthesize $Y_{3-x}Na_xAl_5O_{12}$, $Y_{3-x}K_xAl_5O_{12}$, $Y_3Al_{5-y}V_yO_{12}$, and $Y_{3-x}Na_xAl_{5-y}V_yO_{12}$ garnets with various stoichiometry by an aqueous sol-gel method and study them using X-ray diffraction (XRD) analysis, FTIR spectroscopy, scanning electron microscopy (SEM) and X-ray energy dispersive spectrometry (EDX).

## 2. Results

### 2.1. XRD Analysis

The phase purity and compositional changes of the YAG doped with different amounts of sodium, potassium, and vanadium were controlled by XRD analysis. The X-ray diffraction patterns of the $Y_{3-x}Na_xAl_5O_{12}$ ($0.01 \leq x \leq 2$) samples synthesized at 1000 °C are presented in Figure 1a. Consequently, all the diffraction peaks in the XRD patterns of $Y_{3-x}Na_xAl_5O_{12}$ products with x = 0.01, 0.05, 0.1, 0.15, 0.3, 0.5, and 1.0 match very well to the standard XRD data of pure YAG [PDF #96-152-9038]. All high-intensity peaks are identified and attributed to the characteristic cubic garnet lattice. However, the XRD patterns of $Y_{3-x}Na_xAl_5O_{12}$ with higher introduced amounts of sodium (x = 1.5 and 2) revealed that the perovskite $YAlO_3$ ($2\theta \approx 24.4°$, 34.6°, 45.8°) and aluminum oxide $Al_2O_3$ ($2\theta \approx 30.2°$, 58°, 66.7°) impurity phases appeared along with the dominant YAG phase. The intensity and amount of the peaks of impurity phases increased with increasing the molar part of sodium in the compounds. As a result, the XRD pattern of $Y_1Na_2Al_5O_{12}$ contains more peaks with higher intensities of impurity phases than the XRD pattern of $Y_{1.5}Na_{1.5}Al_5O_{12}$. Thus, the monophasic $Y_{3-x}Na_xAl_5O_{12}$ garnets can be successfully obtained when the introduced amount of sodium is in the range of $0.01 \leq x \leq 1$.

Figure 1b shows the XRD patterns of the $Y_{3-x}K_xAl_5O_{12}$ samples sintered at 1000 °C with the different substitutional levels of K for Y ($0.01 \leq x \leq 1$) in the compounds. As can be seen, the obtained XRD patterns of $Y_{3-x}K_xAl_5O_{12}$ products with x = 0.01, 0.05, 0.1, 0.15, 0.3, and 0.5 are in good agreement with the reference data [PDF #96-152-9038] and correspond to the single-phase YAG, i.e., all diffraction lines are attributed to the garnet phase. When the introduced amount of potassium is x = 1, the YAG is still the major phase; however, the minor amount of the intermediate phase of $YAlO_3$ (YAP) [PDF #96-153-3070] has formed (reflections at $2\theta \approx 16.8°$, 24.5°, 34.15°, 44.7°, and 49.5°). According to the XRD results presented in Figure 1b, the range of $0.01 \leq x \leq 0.5$ could be ascribed to the limit of the molar part of potassium in $Y_{3-x}K_xAl_5O_{12}$ for the formation of the monophasic garnet phase.

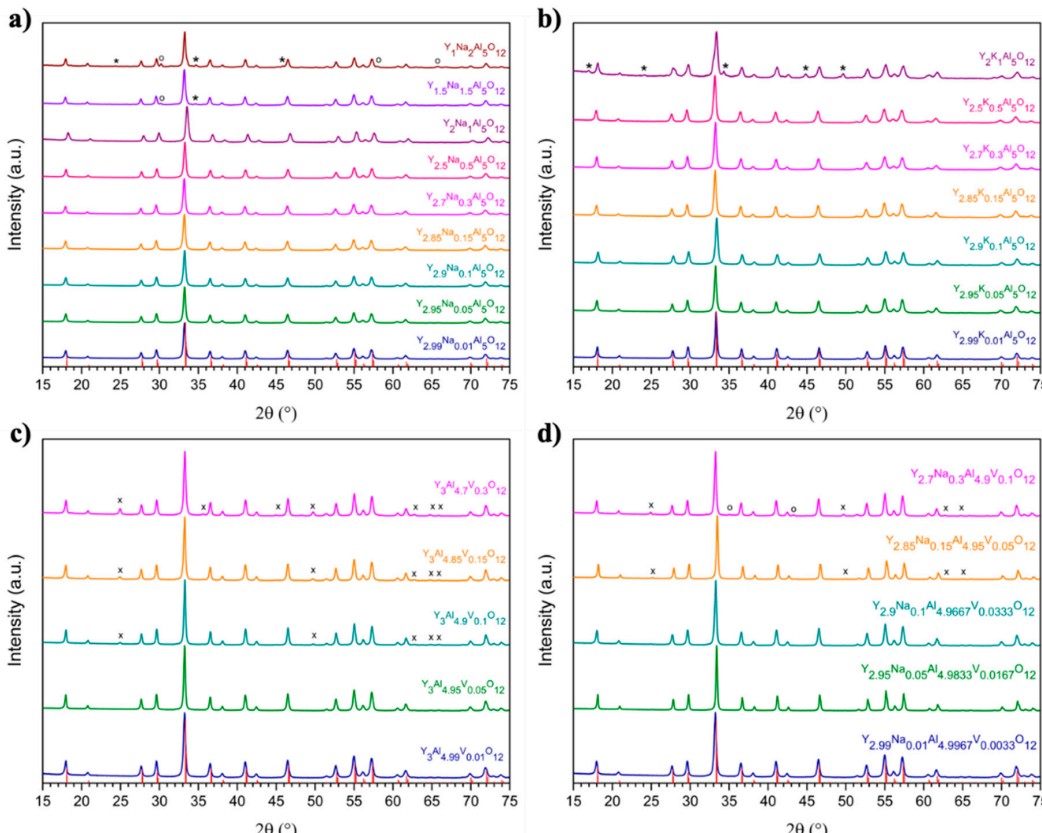

**Figure 1.** XRD diffraction patterns of $Y_{3-x}Na_xAl_5O_{12}$ (**a**), $Y_{3-x}K_xAl_5O_{12}$ (**b**), $Y_3Al_{5-y}V_yO_{12}$ (**c**) and $Y_{3-x}Na_xAl_{5-y}V_yO_{12}$ (**d**) samples synthesized at 1000 °C. The crystalline phases are marked: * - $YAlO_3$ [PDF #96-153-3070], o - $Al_2O_3$ [PDF #96-100-0443], x - $YVO_4$ [PDF #96-901-1138]. Vertical red lines – standard $Y_3Al_5O_{12}$ [PDF #96-152-9038].

The XRD patterns of the $Y_3Al_{5-y}V_yO_{12}$ samples heated at 1000 °C with the different molar parts of vanadium ($0.01 \leq y \leq 3$) in the compounds are presented in Figure 1c. The monophasic garnet structure material is obtained only within a narrow substitutional range of vanadium in $Y_3Al_{5-y}V_yO_{12}$. The measured XRD patterns of $Y_3Al_{4.99}V_{0.01}O_{12}$ and $Y_3Al_{4.95}V_{0.05}O_{12}$ show perfect fit with standard data for pure YAG (PDF #96-152-9038). In Figure 1c, it can be determined that all high intensity peaks correspond to the cubic garnet structure. With increasing the molar part of vanadium to y = 0.1, 0.15, and 0.3 in the $Y_3Al_{5-y}V_yO_{12}$ composition, YAG is still the main phase, but new peaks characteristic of the crystalline impurity phase appear. The peaks at $2\theta \approx 24.9°$, $35.6°$, $45°$, $48.2°$, $49.7°$, $62.6°$, and $64.7°$ are attributed to the yttrium orthovanadate $YVO_4$ [PDF #96-901-1138]. Therefore, the single phase $Y_3Al_{5-y}V_yO_{12}$ garnets can be successfully obtained when the introduced amount of vanadium is in the range of $0.01 \leq y \leq 0.05$.

Figure 1d shows the XRD patterns of the co-doped $Y_{3-x}Na_xAl_{5-y}V_yO_{12}$ samples synthesized at 1000 °C with different substitutional levels of $Na^+$ for $Y^{3+}$ ($0.01 \leq x \leq 0.3$) and $V^{5+}$ for $Al^{3+}$ ($0.0033 \leq y \leq 0.1$) in the compounds. The XRD results revealed that $Y_{2.99}Na_{0.01}Al_{4.9967}V_{0.0033}O_{12}$, $Y_{2.95}Na_{0.05}Al_{4.9833}V_{0.0167}O_{12}$, and $Y_{2.9}Na_{0.1}Al_{4.9667}V_{0.0333}O_{12}$ ceramics are crystalline monophasic compounds having a garnet structure, i.e., all observed diffraction peaks match very well to the standard XRD data of pure YAG (PDF #96-152-9038). As observed in the XRD patterns of $Y_{2.85}Na_{0.15}Al_{4.95}V_{0.05}O_{12}$ and $Y_{2.7}Na_{0.3}Al_{4.9}V_{0.1}O_{12}$ samples, besides the main YAG phase, additional peaks ($2\theta \approx 25°$, $49.8°$, $62.7°$, $65°$) of the foreign $YVO_4$ [PDF #96-901-1138] phase were also detected. Furthermore, in the XRD pattern of the $Y_{2.7}Na_{0.3}Al_{4.9}V_{0.1}O_{12}$ compound, there are two additional peaks at $2\theta \approx 35.1°$ and $43.3°$ which belong to the $Al_2O_3$ [PDF #96-100-0443] phase. Thus, we can

conclude that the ranges of $0.01 \leq x \leq 0.1$ and $0.0033 \leq y \leq 0.0333$ could be attributed to the limits of the sodium and vanadium molar parts in co-substituted $Y_{3-x}Na_xAl_{5-y}V_yO_{12}$.

## 2.2. FTIR Analysis

The FTIR spectroscopy was used as an additional tool for the structural characterization of substituted garnets synthesized at 1000 °C. In addition, the FTIR spectra of $Y_{3-x}Na_xAl_5O_{12}$, $Y_{3-x}K_xAl_5O_{12}$, $Y_3Al_{5-y}V_yO_{12}$, and $Y_{3-x}Na_xAl_{5-y}V_yO_{12}$ samples are shown in Figure 2. The most important feature of all obtained FTIR spectra is that the broad band in the range from 900 to 400 $cm^{-1}$ is split into several smaller peaks (~425 $cm^{-1}$, ~450 $cm^{-1}$, ~564 $cm^{-1}$, ~686 $cm^{-1}$, ~720 $cm^{-1}$, ~784 $cm^{-1}$). This feature of the IR spectrum, according to the literature, is attributed to the metal-oxygen (M-O) vibrations of the isolated $(AlO_4)$ tetrahedral and $(AlO_6)$ octahedral units in the garnet structure, i.e., proves the formation of garnet structure [2,3,29]. It is interesting to note, that $Y_{1.5}Na_{1.5}Al_5O_{12}$, $Y_1Na_2Al_5O_{12}$, $Y_2K_1Al_5O_{12}$, $Y_3Al_{4.9}V_{0.1}O_{12}$, $Y_3Al_{4.85}V_{0.15}O_{12}$, $Y_3Al_{4.7}V_{0.3}O_{12}$, $Y_{2.85}Na_{0.15}Al_{4.95}V_{0.05}O_{12}$, and $Y_{2.7}Na_{0.3}Al_{4.9}V_{0.1}O_{12}$ are multiphasic samples, as we know from XRD results, but the characteristic band splitting was also observed. It confirms the fact that in all garnets, the dominant crystalline phase is the YAG phase. A weak intensity absorption band located at ~2350 $cm^{-1}$ corresponds to the carbon dioxide adsorbed from the atmosphere [2,3,29]. However, the bands at ~3400–3300 $cm^{-1}$ and ~1600 $cm^{-1}$ due to the O–H vibration of $H_2O$ absorbed by the samples can be observed in some FTIR spectra [29]. Consequently, the FTIR results are consistent with the XRD results and prove that, in all cases, garnet-type materials are formed.

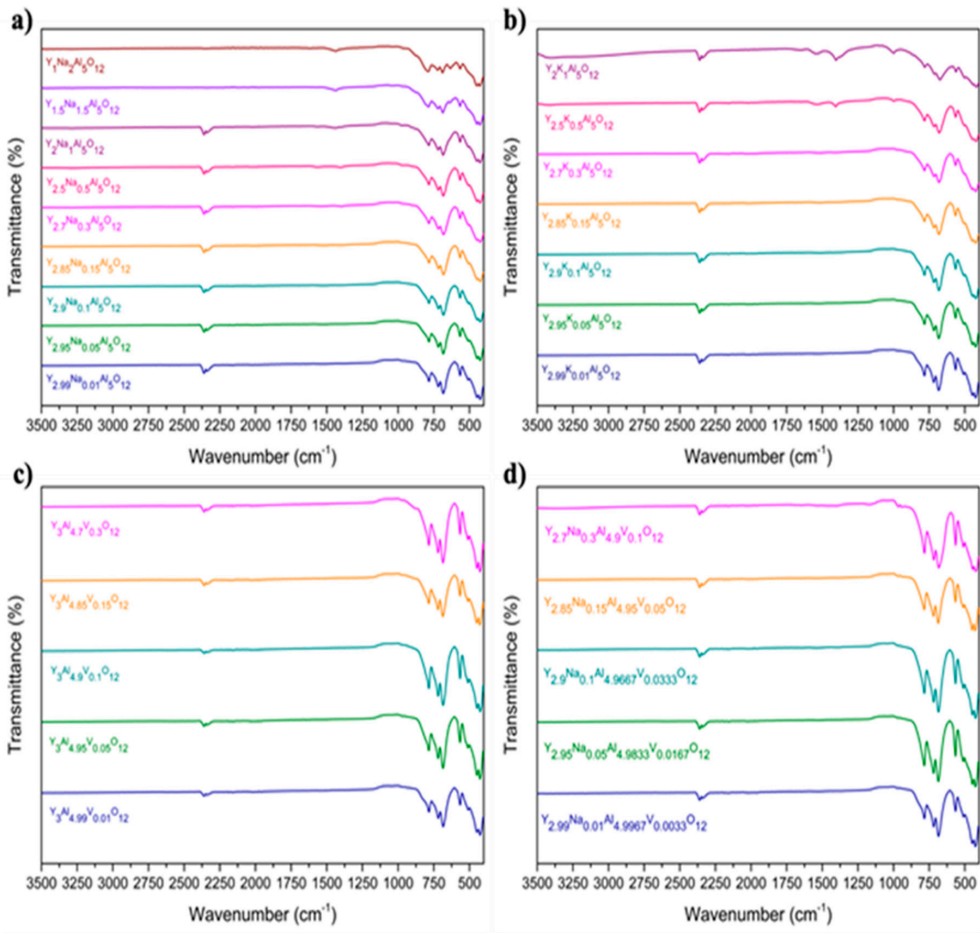

**Figure 2.** FTIR spectra of the $Y_{3-x}Na_xAl_5O_{12}$ (**a**), $Y_{3-x}K_xAl_5O_{12}$ (**b**), $Y_3Al_{5-y}V_yO_{12}$ (**c**), and $Y_{3-x}Na_xAl_{5-y}V_yO_{12}$ (**d**) samples synthesized at 1000 °C for 2 h.

### 2.3. SEM Analysis

The surface morphology of $Y_{3-x}Na_xAl_5O_{12}$, $Y_{3-x}K_xAl_5O_{12}$, $Y_3Al_{5-y}V_yO_{12}$, and $Y_{3-x}Na_xAl_{5-y}V_yO_{12}$ garnets prepared by the sol-gel method was studied with a scanning electron microscope (SEM). It is important to note that, for all obtained samples, the main morphological features are almost identical. In Figure 3, the SEM micrographs of the representative samples of $Y_{3-x}Na_xAl_5O_{12}$, $Y_{3-x}K_xAl_5O_{12}$, $Y_3Al_{5-y}V_yO_{12}$, and $Y_{3-x}Na_xAl_{5-y}V_yO_{12}$ are shown. We can clearly see that the products are homogeneous, well dispersed, and have a very porous structure. The origin of pores can be explained by escaping gases during the firing process. Moreover, the particles are of irregular sphere-like shape and are formed from agglomerated primary crystallites. The average crystallite size of $Y_{2.9}Na_{0.1}Al_5O_{12}$, $Y_{2.9}K_{0.1}Al_5O_{12}$, $Y_3Al_{4.99}V_{0.01}O_{12}$, and $Y_{2.99}Na_{0.01}Al_{4.9967}V_{0.0033}O_{12}$ specimens is 43 nm, 41 nm, 39 nm, and 40 nm, respectively. Furthermore, according to the literature data, agglomeration and porous structure are the characteristic textural features for garnet-type materials synthesized by sol-gel synthesis [2,16,21].

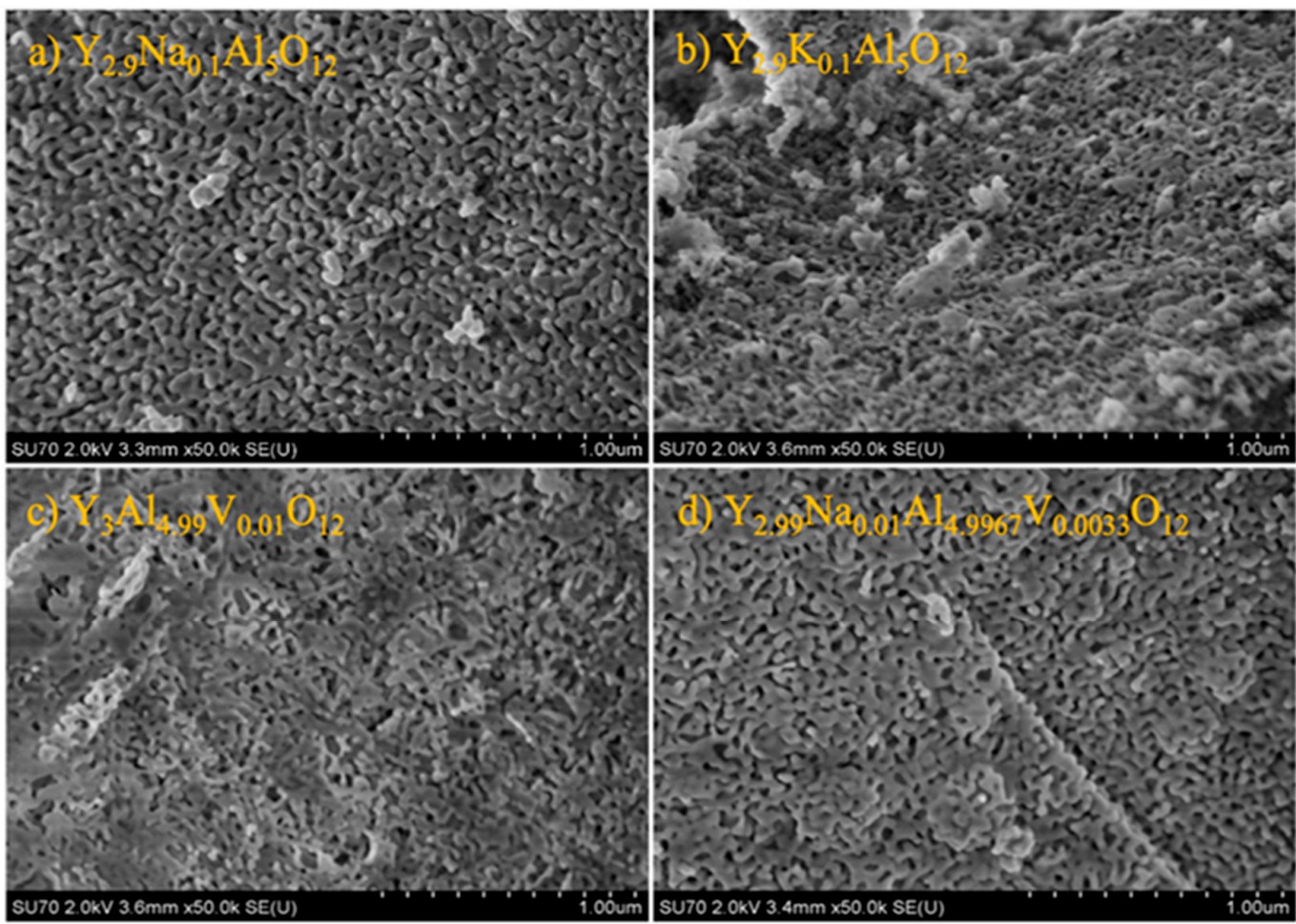

**Figure 3.** SEM micrographs of the $Y_{2.9}Na_{0.1}Al_5O_{12}$ (**a**), $Y_{2.9}K_{0.1}Al_5O_{12}$ (**b**), $Y_3Al_{4.99}V_{0.01}O_{12}$ (**c**), and $Y_{2.99}Na_{0.01}Al_{4.9967}V_{0.0033}O_{12}$ (**d**) garnets synthesized at 1000 °C.

### 2.4. EDX Analysis

The chemical composition of differently doped YAG has been investigated by EDX analysis. In the EDX spectra of sodium-substituted $Y_{3-x}Na_xAl_5O_{12}$ garnets, the characteristic lines for yttrium (Y), aluminum (Al), sodium (Na), and oxygen (O) are observed. The molar ratio of elements corresponds to the nominal chemical composition of the compounds. The EDX spectra of all of the sodium-containing garnets were similar, and, therefore, Figure 4 shows the representative EDX spectra of two $Y_{2.7}Na_{0.3}Al_5O_{12}$ (a) and

$Y_2Na_1Al_5O_{12}$ (b) samples. As it is observed, the intensity of the characteristic lines of sodium and yttrium is directly related to their molar part in the compound. For example, when 0.3 mol of Na was introduced, the intensity of the characteristic line was low, but when 1 mol of Na was introduced, the intensity increased significantly.

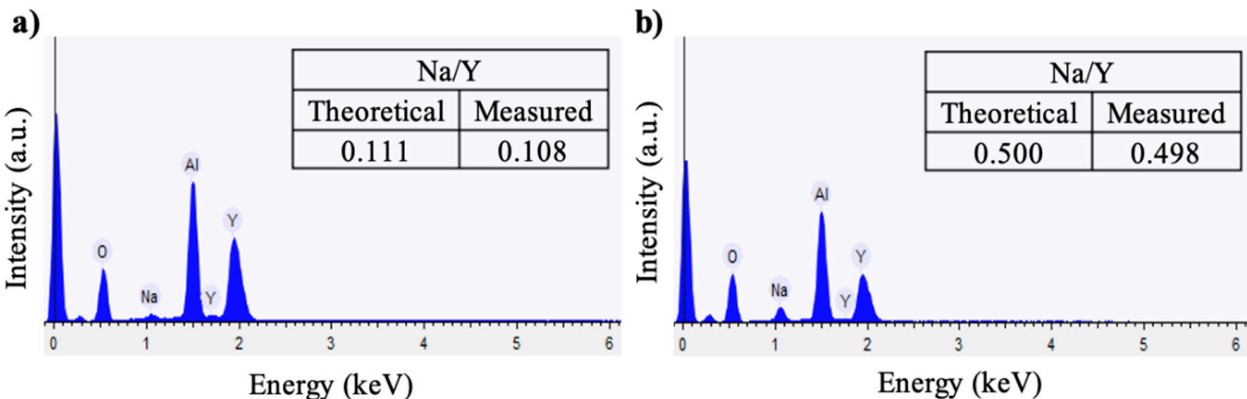

**Figure 4.** EDX analysis spectra of $Y_{2.7}Na_{0.3}Al_5O_{12}$ (**a**) and $Y_2Na_1Al_5O_{12}$ (**b**) samples.

The EDX spectra of potassium-substituted $Y_{3-x}K_xAl_5O_{12}$ garnets confirm the presence of yttrium (Y), aluminum (Al), potassium (K), and oxygen (O) elements in the compounds. The EDX spectra of potassium-containing garnet specimens were also almost identical in the main features. Figure 5 presents the EDX spectra of two $Y_{2.85}K_{0.15}Al_5O_{12}$ (a) and $Y_2K_1Al_5O_{12}$ (b) samples. Again, the intensity of the line of potassium increases with an increasing molar part of potassium. At the same time, the intensity of the line of yttrium decreases as its molar part decreases. Apparently, the intensity of these lines correctly depends on the molar ratio of elements in compounds. Thus, these results are in good agreement with the nominal composition of precursors.

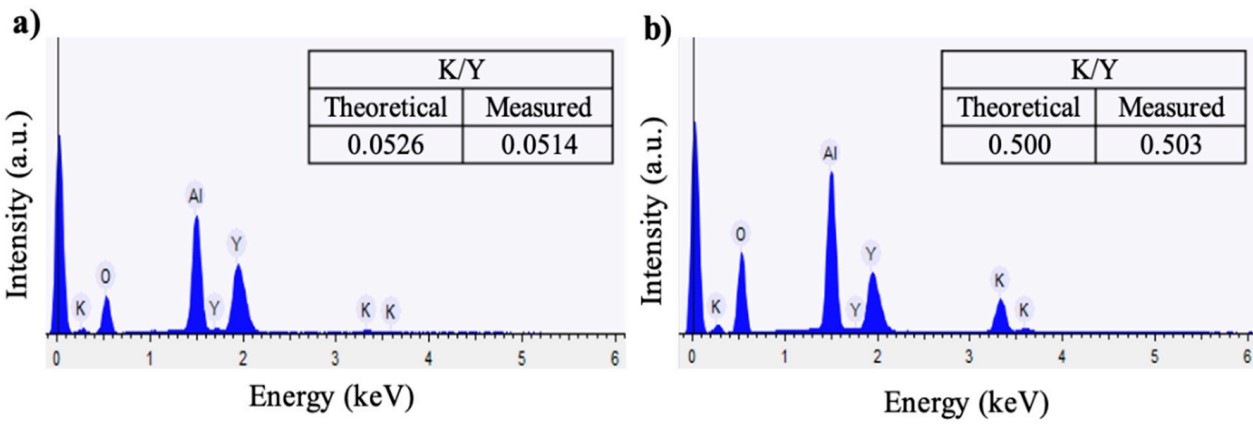

**Figure 5.** EDX analysis spectra of $Y_{2.85}K_{0.15}Al_5O_{12}$ (**a**) and $Y_2K_1Al_5O_{12}$ (**b**) samples.

The EDX spectra of representative samples of $Y_3Al_{5-y}V_yO_{12}$ and $Y_{3-x}Na_xAl_{5-y}V_yO_{12}$ garnets ($Y_3Al_{4.9}V_{0.1}O_{12}$ (a) and $Y_{2.9}Na_{0.1}Al_{4.9667}V_{0.0333}O_{12}$ (b), respectively) are shown in Figure 6. All EDX spectra of $Y_3Al_{5-y}V_yO_{12}$ compounds contain characteristic lines of yttrium (Y), aluminum (Al), vanadium (V), and oxygen (O), and they correspond to the nominal chemical compositions. In the case of $Y_{3-x}Na_xAl_{5-y}V_yO_{12}$ garnets, the EDX spectra confirm the presence of Y, Al, Na, V, and O elements in the compounds. In addition, the intensity of the characteristic element lines is directly related to their molar parts in the compound.

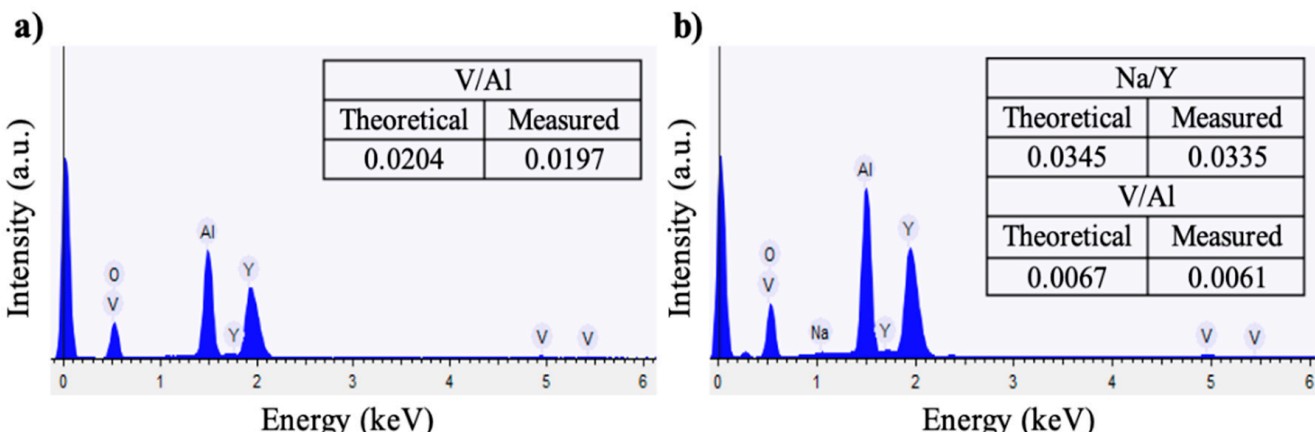

**Figure 6.** EDX analysis spectra of $Y_3Al_{4.9}V_{0.1}O_{12}$ (**a**) and $Y_{2.9}Na_{0.1}Al_{4.9667}V_{0.0333}O_{12}$ (**b**) samples.

Finally, in all cases, the obtained average atomic percentage of the characteristic elements corresponds to the stoichiometric composition of the synthesized garnet. Furthermore, in the tables inserted in Figures 4–6, the calculated theoretical and measured Na/Y, K/Y, V/Al, Na/Y, and V/Al atomic ratios for the representative samples of $Y_{3-x}Na_xAl_5O_{12}$, $Y_{3-x}K_xAl_5O_{12}$, $Y_3Al_{5-y}V_yO_{12}$, $Y_{3-x}Na_xAl_{5-y}V_yO_{12}$, are given, respectively. We can clearly see that in all instances, the measured atomic ratio values are in very good agreement with the theoretical values. Thus, the EDX results match very well with the nominal composition of the precursors and show that the elemental distribution in the samples is homogeneous.

## 3. Experimental

### 3.1. Materials

The Yttrium nitrate hexahydrate ($Y(NO_3)_3 \cdot 6H_2O$, Alfa Aesar, 99.9%), aluminum nitrate nonahydrate ($Al(NO_3)_3 \cdot 9H_2O$, Carl Roth, 98%), sodium acetate ($CH_3COONa$, Carl Roth, 99%), potassium acetate ($CH_3COOK$, Carl Roth, 99%), ammonium monovanadate ($NH_4VO_3$, Carl Roth, 99.8%), and trisodium monovanadate ($Na_3VO_4$, Carl Roth, 99%) were used as starting materials and citric acid ($C_6H_8O_7$, Chempur, 99.7%) was used as a complexing agent.

### 3.2. Synthesis

The $Y_{3-x}Na_xAl_5O_{12}$ ($0.01 \leq x \leq 2$), $Y_{3-x}K_xAl_5O_{12}$ ($0.01 \leq x \leq 1$), $Y_3Al_{5-y}V_yO_{12}$ ($0.01 \leq y \leq 3$), and $Y_{3-x}Na_xAl_{5-y}V_yO_{12}$ ($0.01 \leq x \leq 0.3$; $0.0033 \leq y \leq 0.1$) garnet samples were synthesized by an aqueous sol-gel method. First, stoichiometric amounts of starting materials were dissolved in 50 mL of distilled water. The transparent solutions of metal salts were obtained after stirring at 80 °C for 2 h in beakers covered with a watchglass. Next, the complexing reagent, citric acid (molar ratio metals:citric acid = 1:1), was added to this solution with continuous stirring for 24 h in covered beakers. Then, the resulting solution was concentrated by slow evaporation at 200 °C in an open beaker under stirring until the transparent sol-gel turned into a transparent sticky gel. The gels were dried in an oven at 150 °C for 10 h. The obtained xerogels were ground in an agate mortar and heated in air at 1000 °C for 2 h. The heating rate was 5 °C/min. Finally, the synthesized garnets were ground in an agate mortar. The simplified scheme of the sol-gel route used is presented in Figure 7.

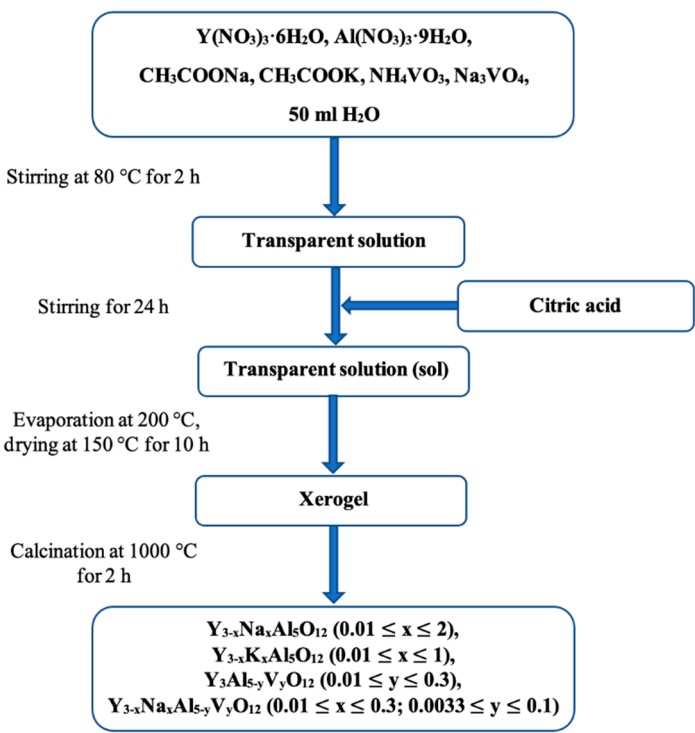

**Figure 7.** Scheme of sol-gel synthesis of garnets with different stoichiometry.

### 3.3. Characterization

The phase purity and crystallinity of synthesized samples were characterized by X-ray powder diffraction (XRD) analysis on a Rigaku MiniFlex II diffractometer (Rigaku Corporation, Tokyo, Japan) using Cu K$\alpha$1 radiation. The infrared (IR) spectroscopy measurements were conducted using a Bruker Alpha FTIR spectrometer (Billerica, MA, USA). The scanning electron microscope (SEM) images were taken using a Hitachi SU-70 SEM (FE-SEM, Hitachi, Tokyo, Japan). The particle size measurements were done using open-source Fiji (ImageJ) software by accidently selecting random particles. The elemental composition of synthesized specimens was determined by X-ray energy dispersive spectrometry (EDX) on a Hitachi TM3000 SEM with an EDX detector.

### 4. Conclusions

In this study, for the first time to the best of our knowledge, the new $Y_{3-x}Na_xAl_5O_{12}$, $Y_{3-x}K_xAl_5O_{12}$, $Y_3Al_{5-y}V_yO_{12}$, and $Y_{3-x}Na_xAl_{5-y}V_yO_{12}$ garnets with different stoichiometry were successfully synthesized by the aqueous sol-gel method. Multiple characterization techniques were employed to study the phase purity, morphological features, and composition of the end products. In addition, from the results of XRD analysis, it was determined that monophasic products could be obtained only with a limited amount of the element used for substitution. For example, the substitution of yttrium by sodium with the formation of monophasic $Y_{3-x}Na_xAl_5O_{12}$ was possible in the cases with $0.01 \leq x \leq 1$. The monophasic $Y_{3-x}K_xAl_5O_{12}$ garnets were obtained when the introduced amount of potassium was in the range of $0.01 \leq x \leq 0.5$. The narrow range of $0.01 \leq y \leq 0.05$ can be ascribed to the limit of the molar part of vanadium in $Y_3Al_{5-y}V_yO_{12}$. The single-phase co-substituted $Y_{3-x}Na_xAl_{5-y}V_yO_{12}$ was formed when the introduced amounts of sodium and vanadium were $0.01 \leq x \leq 0.1$ and $0.0033 \leq y \leq 0.0333$, respectively. Furthermore, the obtained FTIR spectroscopy results were in good agreement with the XRD analysis data and confirmed that YAG was the main crystalline phase in all products. The SEM micrographs showed that the samples were homogeneous with a porous surface and composed of nano-sized agglomerated crystallites. Moreover, the EDX analysis results revealed that the real stoichiometry of products is very similar to the nominal stoichiometry. Therefore,

according to this study, rare earth or transition metal-doped garnets with novel chemical compositions can be used as optical host materials.

**Author Contributions:** Conceptualization, A.L. and A.Z.; methodology, D.V.; software, A.M.d.L.; formal analysis, D.V. and S.P.; investigation, D.V., A.L. and S.P.; data curation, A.Z.; writing—original draft preparation, D.V.; writing—review and editing, A.K.; visualization, A.M.d.L.; supervision, A.K.; project administration, A.K.; funding acquisition, A.Z. All authors have read and agreed to the published version of the manuscript.

**Funding:** This research received no external funding.

**Data Availability Statement:** Not applicable.

**Conflicts of Interest:** The authors declare no conflict of interest.

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
