# Peer review of "Sol-Gel Synthesis and Characterization of Novel Y3−xMxAl5−yVyO12 (M—Na, K) Garnet-Type Compounds"

_inorganics, doi:10.3390/inorganics11020058_

Round 1

Reviewer 1 Report

My comments are in attachement.

Author Response

In this my letter I enclosed the revised according to the Reviewer 1 comments our manuscript ”Sol-gel synthesis and characterization of the novel and Y3-xMxAl5-yVyO12  (M – Na, K) garnet-type compounds” by D. Vistorskaja, A. Laurikenas, A. Montejo de Luna, A. Zarkov, S. Pazylbek, A. Kareiva. (Manuscript No.: inorganics-2094399).

First of all, we would like to thank Reviewer for the valuable remarks. We made almost all changes suggested by Reviewer. The changes made are outlined in the revised manuscript.

We were not able to find the article “ACS Applied Materials and Interfaces 25345-26243”, since the publication year are not indicated. However, the additional 3 articles by the same authors are cited in the revised manuscript.

Also, since card number for X-ray diffraction data are indicated, we think that the citation of related original paper would be redundant information. At least, the most of the authors, as well as the Journals avoid double citation.

I hope very much that the revised manuscript will be suitable for publication in Inorganics.

I am looking forward to hearing from you.

Sincerely yours,

Prof. Aivaras Kareiva

Reviewer 2 Report

Authors prepared different stoichiometric ratios of some elements doped in YAG by sol gel method. Personally I couldn't find any scientific achievement for that work. The sol gel is very well known method and authors didn't show what is the interest of doping of such element and what is the purpose of this work!. Is it really difficult to be prepared? what is the challenges? For me, the study doesn't offer any scientific significance to be published. Therefore I recommend the rejection.

Author Response

In this my letter I enclosed the revised according to the Reviewer 2 comments our manuscript ”Sol-gel synthesis and characterization of the novel and Y3-xMxAl5-yVyO12  (M – Na, K) garnet-type compounds” by D. Vistorskaja, A. Laurikenas, A. Montejo de Luna, A. Zarkov, S. Pazylbek, A. Kareiva. (Manuscript No.: inorganics-2094399).

Rev 2

Authors prepared different stoichiometric ratios of some elements doped in YAG by sol gel method. Personally I couldn't find any scientific achievement for that work. The sol gel is very well known method and authors didn't show what is the interest of doping of such element and what is the purpose of this work!. Is it really difficult to be prepared? what is the challenges? For me, the study doesn't offer any scientific significance to be published. Therefore I recommend the rejection.

Answer

In this study, for the first time the new Y3-xNaxAl5O12,
Y3-xKxAl5O12, Y3Al5-yVyO12, Y3-xNaxAl5-yVyO12 garnets with different stoichiometric were successfully synthesized by the aqueous sol-gel method. Unfortunately, solid state reaction method failed to obtain these garnets at such low temperature. The rare earth or transition metals doped garnets with novel chemical compositions will be used as optical host materials. Thus, we expect some enhanced luminescent properties from these new compounds.

I hope very much that the revised manuscript will be suitable for publication in Inorganics.

I am looking forward to hearing from you.

Sincerely yours,

Prof. Aivaras Kareiva

Reviewer 3 Report

The paper reported the synthesis of alkali metal doped and transition metal doped YAG via a sol-gel method and the crystallographic/optical/ morphological characterizations of the synthesized products. The presented data support the conclusions drawn by the authors.

- I suggest Figure 2 could be improved to show the small impurity peaks at high doping concentrations.

- Could authors explain more about the importance of the substitution of Na, K, and V in YAG? What properties (optical, mechanical, etc) are expected to be improved by doping?

Author Response

In this my letter I enclosed the revised according to the Reviewer’s comments our manuscript ”Sol-gel synthesis and characterization of the novel and Y3-xMxAl5-yVyO12  (M – Na, K) garnet-type compounds” by D. Vistorskaja, A. Laurikenas, A. Montejo de Luna, A. Zarkov, S. Pazylbek, A. Kareiva. (Manuscript No.: inorganics-2094399).

First of all, we would like to thank Reviewer 3 for the valuable remarks.

  • Comment

- I suggest Figure 2 could be improved to show the small impurity peaks at high doping concentrations.

Answer:

The impurity peaks in the XRD patterns of samples with high doping concentrations are marked. The phases are described in Figure caption.

  • Comment

- Could authors explain more about the importance of the substitution of Na, K, and V in YAG? What properties (optical, mechanical, etc) are expected to be improved by doping?

Answer:

In this study, for the first time the new Y3-xNaxAl5O12,
Y3-xKxAl5O12, Y3Al5-yVyO12, Y3-xNaxAl5-yVyO12 garnets with different stoichiometric were successfully synthesized by the aqueous sol-gel method. Thus, the substitution of Na, K, and V in YAG was not investigated before. It is well known, that luminescence properties of different phosphors depend very much on chemical composition of matrix. We expect, that by doping with rare earth or transition metals, these doped garnets with novel chemical compositions will show different optical properties in comparison with un-doped YAG. Thus, we expect some enhanced luminescent properties from these new compounds which could be explore for the fabrication of new LEDs.

I hope very much that the revised manuscript will be suitable for publication in Inorganics.

I am looking forward to hearing from you.

Sincerely yours,

Prof. Aivaras Kareiva